# The effect of cooperator recognition on competition among clones in spatially structured microbial communities

**Adrienna Bingham[1¤a], Aparajita Sur [2¤b], Leah B. Shaw[2], Helen A. Murphy [3]***

**1** Department of Applied Science, William & Mary, Williamsburg, VA, United States of America, **2** Department of Mathematics, William & Mary, Williamsburg, VA, United States of America, **3** Department of Biology, William & Mary, Williamsburg, VA, United States of America

¤a Current address: Sensors & Electromagnetic Application Laboratory, Georgia Tech Research Institute, Smyrna, GA, United States of America
¤b Current address: Division of Biostatistics, University of Minnesota, Minneapolis, MN, United States of America

* hamurphy@wm.edu

**Data Availability Statement:** Code to generate and analyze colonies is available on GitHub (https://github.com/abingham3/SpatialStructureSim). The summary output for simulations performed for this

## Abstract

In spatially structured microbial communities, clonal growth of stationary cells passively generates clusters of related individuals. This can lead to stable cooperation without the need for recognition mechanisms. However, recent research suggests that some biofilm-forming microbes may have mechanisms of kin recognition. To explore this unexpected observation, we studied the effects of different types of cooperation in a microbial colony using spatially explicit, agent-based simulations of two interacting strains. We found scenarios that favor a form of kin recognition in spatially structured microbial communities. In the presence of a "cheater" strain, a strain with greenbeard cooperation was able to increase in frequency more than a strain with obligate cooperation. This effect was most noticeable in high density colonies and when the cooperators were not as abundant as the cheaters. We also studied whether a polychromatic greenbeard, in which cells only cooperate with their own type, could provide a numerical benefit beyond a simple, binary greenbeard. We found the greatest benefit to a polychromatic greenbeard when cooperation is highly effective. These results suggest that in some ecological scenarios, recognition mechanisms may be beneficial even in spatially structured communities.

## Introduction

Microbial life is rife with complex interactions between and within species. As a necessary part of existence, microbial social interactions can vary from chemical warfare, to competition, to synchronization, to cooperation [1, 2]. Like all cooperative communities, microbes are susceptible to invasion by selfish individuals who benefit from cooperation [3], but do not contribute [4–10]. Despite the potential for invasion, cooperative behaviors are prevalent in natural systems. Such behaviors can often be explained through kin selection: individuals cooperate with

study are included as supplementary files. The raw data for this paper (large files of simulated colonies) are available upon request.

**Funding:** The research was funded by a Jeffress Trust for Interdisciplinary Research Award to HAM, as well as National Science Foundation grant DEB-1839555 to HAM and DMS-1715651 to LBS. The funders had no role in study design, data collection and analysis, decision to publish, or preparation of the manuscript.

**Competing interests:** The authors have declared that no competing interests exist.

relatives that likely share genes for the behavior. If the fitness benefits of the behavior are large enough, the frequency of the cooperative allele(s) will increase in the population, such that cooperation will outcompete other strategies [11, 12].

One way to ensure the stability of kin-selected cooperation is through kin recognition in which individuals recognize and preferentially cooperate with related individuals [13]. While recognition of relatives has been documented in animals, it is uncommon in microbial systems. Instead, there is "kind" recognition where cooperators recognize each other through a particular signal; this system is known as a greenbeard. A greenbeard locus encodes a cooperative behavior (a single or multiple, tightly linked genes), a signal of cooperation, and the ability to recognize the signal in others [14, 15]. For a greenbeard system to function, there must be relatedness at the locus, but not necessarily across the entire genome. Microbial examples include cell adhesion proteins that allow cells to attach to one another, as has been reported in yeast and social amoebae [16–19].

In aggregative cooperative behaviors that require motile microbes to locate one another before adhering, such as the formation of foraging slugs, fruiting bodies, or swarms, rare instances of kin- and self-recognition have been reported [9, 20, 21]. Rather than a binary system of cooperation or non-cooperation, as is expected for a greenbeard locus, the proteins responsible for cellular adhesion exhibit a spectrum of cooperation (e.g., strength of cell-to-cell interaction) related to allelic variation, thus creating a "polychromatic greenbeard" [22, 23]. A polychromatic greenbeard is more similar to traditional kin recognition, as individuals with identical "shades" of green tend to be more closely related across the genome, as long as rates of recombination are low [24, 25].

In contrast to motile microbes, many microbes live and interact in stationary, spatially-structured communities known as biofilms. These communities are characterized by attachment to a surface, an extracellular matrix produced by the microbes, cellular differentiation, and increased resistance to environmental stressors [26]. Biofilms exist in nearly every environment in which microbes can be found, including medical settings where their presence is a particular risk due to their increased resistance to antimicrobials [27, 28]. Biofilms require individuals to cooperate and produce goods that will be used by other members of the community (i.e., extracellular matrix, drug efflux pumps, diffusible enzymes, quorum sensing molecules) [2]. Mathematical modeling, simulations, and microbial experiments have demonstrated that unlike in aggregative and motile phenotypes, biofilms do not require a recognition system for cooperation to be stable (reviewed in [29]). Clonal growth generates local patches of high-relatedness and leads to lineage sorting [30–32]. In this way, passive spatial assortment leads to kin selection without the need for kin recognition. Despite the body of work suggesting recognition systems are not required for stable cooperation in biofilms, recent research has provided evidence that a cell adhesion protein may actually act in recognition in the budding yeast *Saccharomyces cerevisiae*, a stationary, biofilm-forming eukaryotic microbe [33, 34].

Given this unexpected observation, here we ask whether greenbeard and polychromatic greenbeard systems could provide a benefit to cooperators in stationary microbial communities. While such recognition systems may not be required for stable cooperation, it is possible they provide enough of a fitness benefit to explain the presence of recognition systems in species whose characteristics preclude motility. We hypothesize that even with the passive emergence of clusters of direct kin, the ability to restrict cooperative benefits will lead to increased representation of cooperators in a growing spatially-structured community.

Our approach to addressing this question uses spatially-explicit, agent-based simulations of a growing microbial community containing two cell types. The simulations represent a simplified microbial community in which cells interact under different scenarios (with different sets

of rules). Briefly, in our simulations, cooperators have a slower basal growth rate than non-cooperators, which represents the cost of producing non-diffusible/locally diffused public goods and other traits associated with biofilm formation. Cells that are adjacent to cooperators have an increased growth rate due to the benefit provided by the cooperative goods. A green-beard recognition system is implemented by allowing cooperator cells to only increase the growth rate of cooperator neighbors. Finally, we investigate the effect of a polychromatic greenbeard by simulating microbial communities with two cooperator cell types that have different basal growth rates and can restrict growth benefit to their own cell type. We simulate different scenarios over a range of starting ratios of each cell type and of the strength of cooperation.

## Methods

We used an agent-based model to study the interactions between two cell types in a microbial community. The biological details were inspired by the yeast *Saccharomyces cerevisiae*, in which the potential for a recognition system has been reported [33, 34]. We used a published framework [35], which we adjusted to apply to our research questions. The published model considered the effect two different cell types would have on one another, but did not consider the effect each cell type would have on other cells of its own kind. Our model is summarized in this section and the details of the simulation are given in the S1 File. Briefly, to begin the simulation, cells were placed on a three-dimensional cubic array, and then stochastic Monte Carlo simulation was used to model their division. Social interactions were implemented by altering the rate at which a cell divided based on the occupancy and composition of the immediate neighborhood. When a cell divided, a daughter cell was placed at a nearby location. At the end of the simulation, the number and placement of each cell type was used for calculation.

Each cell type started with a baseline growth rate, $r_{S0}$ and $r_{F0}$. We assumed $r_{S0} < r_{F0}$, or that there is a slower (*S*) and faster (*F*) growing cell type. For example, when only one cell type was capable of cooperation, we designated the cooperators *S* and the non-cooperators *F* to account for the costs of cooperation. In the presence of social interactions, the growth rates for the two cell types were:

$$r_S = (r_{S0} + r_{SF}\phi_F + r_{SS}\phi_S)[1 - \chi(\phi_S + \phi_F)], \tag{1}$$

$$r_F = (r_{F0} + r_{FS}\phi_S + r_{FF}\phi_F)[1 - \chi(\phi_S + \phi_F)]. \tag{2}$$

Parameters and variables are listed in Table 1 and are explained below.

To determine the growth rate of an individual cell, we first calculated the local density of each cell type (fraction of sites occupied), $\phi_S$ and $\phi_F$. These were calculated for a cubic neighborhood within an interaction radius $R_i$ surrounding the growing cell. A competition effect, $\chi$, was incorporated and set to 1, such that the final term in the growth rate calculation, $1 - \chi(\phi_S + \phi_F)$, resulted in a zero growth rate when the neighborhood was full. Social interactions other than competition were set by the parameters $r_{ij}$, where each of *i* and *j* was *S* or *F*. The parameter $r_{ij}$ determined the effect that cell type *j* had on type *i*. For example, when strain *F* was a non-cooperator, its presence would have no effect on the slow strain or itself, so $r_{FF} = r_{SF} = 0$. However, when a strain was a cooperator, it would have a beneficial effect on others of its type, so $r_{SS} > 0$.

The growth rate parameters, $r_{ij}$, were varied to simulate different social interactions and included some values that allowed the slower cells to achieve the same or greater growth than the non-cooperative fast cells. We considered four scenarios in our simulations: baseline competition, obligate cooperation, greenbeard cooperation, and polychromatic greenbeard

**Table 1. Simulation variables.**

| Parameter | Meaning/purpose | Value |
|---|---|---|
| $r_S$ | Growth rate of slow cell | Calculated in simulation |
| $r_F$ | Growth rate of fast cell | Calculated in simulation |
| $r_{S0}$ | Baseline growth rate of slow cell | 0.05 |
| $r_{F0}$ | Baseline growth rate of fast cell | 0.1 |
| $r_{SS}$ | Effect of slow cells on other slow cells | 0, 0.1, 0.3, 0.5, 0.7 |
| $r_{FS}$ | Effect of slow cells on fast cells | 0, 0.1, 0.3, 0.5, 0.7 |
| $r_{FF}$ | Effect of fast cells on other fast cells | 0, 0.1, 0.3, 0.5, 0.7 |
| $r_{SF}$ | Effect of fast cells on slow cells | 0, 0.1, 0.3, 0.5, 0.7 |
| $\chi$ | Competition effect | 1 |
| $\phi_S$ | Occupancy fraction of slow cells in neighborhood | Calculated in simulation |
| $\phi_F$ | Occupancy fraction of fast cells in neighborhood | Calculated in simulation |
| $N_c$ | Maximum biofilm length and width | 50 |
| $N_z$ | Maximum biofilm height | 100 |
| $T$ | Maximum number of simulated cells | 50,000 |
| $d$ | Length of inoculation square | 15 |
| $f$ | Fraction of inoculation square filled with cells | 0.05 |
| $f_S$ | Fraction of slow cells in inoculum | 0.1, 0.3, 0.5, 0.7, 0.9 |
| $R_i$ | Interaction radius | 3 |
| $R_d$ | Displacement radius | 3 |
| $\Delta t$ | Time step | 0.1 |

cooperation. First, for baseline competition, all social interactions other than competition were absent ($r_{SS} = r_{FF} = r_{SF} = r_{FS} = 0$). Next, for obligate cooperation, both $r_{SS}$, $r_{FS} > 0$, while $r_{FF}$, $r_{SF} = 0$. We set $r_{FS} = r_{SS}$ so that non-cooperators derived the same benefit from cooperator neighbors as a cooperator would. This was the simple case in which non-cooperative cheaters could benefit from the presence of cooperators. In the third scenario, a greenbeard cooperator, we again set $r_{SS} > 0$. However, the cooperator could now restrict its benefit to its own cell type, and the non-cooperator strain did not benefit from the public goods, so $r_{FS}$, $r_{FF}$, $r_{SF} = 0$. Finally, in the polychromatic greenbeard scenario, both the fast and slow strain were cooperators that were able to restrict their cooperation to their own strain, $r_{SS} = r_{FF} > 0$, and $r_{SF}$, $r_{FS} = 0$. To determine if a polychromatic greenbeard was beneficial, for comparison, we also considered the situation where both the fast and slow cooperators were simple greenbeards that do not restrict cooperation, $r_{SS} = r_{FF} = r_{SF} = r_{FS} > 0$.

The simulations began with a diluted inoculum of cells into a small square of size $d{\times}d$ at the center of the bottom layer of our cubic array (see Fig 1A), similar to a migration event in the environment or a droplet on a petri dish. This allowed us to simulate outward expansion of an initial community/colony from a founding event, in contrast to the simulation in [35] with random inoculation throughout their whole bottom layer. A fraction $f = 0.05$ of the sites were randomly selected to fill with cells, and a fraction $f_S$ of the initial cells were assigned as the cell type with slow baseline growth, with the remainder as the faster type.

When a cell divided, the next step was to find an empty site for the daughter cell to be placed. If there was an empty site within a square neighborhood of radius $R_d$ in the same horizontal plane, the cell divided horizontally, pushing other cells aside as needed (see S1 File). A similar mechanism to push nearby cells was implemented by [35]. While we set the value of $R_d$ to 3, their data suggested that the value could be as high as 5. If there were no empty sites in the displacement neighborhood in the same horizontal plane, the daughter cell was placed

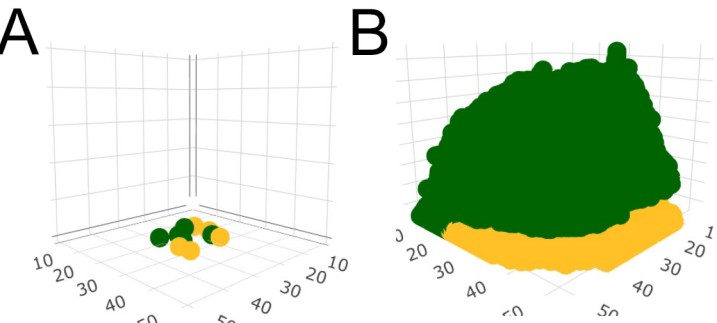

**Fig 1. The simulation.** A: Schematic of cells randomly distributed within a centered square droplet on the bottom surface of the cubic array in the simulation; the size of the cubic array was set to 50 cells long by 50 cells wide by 100 cells high. Axes represent the location in simulated space. The green squares are slow/cooperative cells and the yellow squares are fast/non-cooperative cells. This figure depicts a cell inoculation with $f_S = 0.5$, or an equal proportion of each cell type. B: Schematic of cells at the end of a simulation for which $f_S = 0.5$, $r_{SS} = 0.5$, and $r_{FS}$, $r_{FF}$, $r_{SF} = 0$, a greenbeard cooperator scenario.

directly above the mother cell as this has been shown to be a characteristic of *S. cerevisiae* [35]. If there were cells above the mother cell, they were pushed upwards.

Cell growth was simulated until a specified stopping point was reached. In most cases, the stopping point was when the community reached a total of 50,000 cells (see Fig 1B), which, through multiple simulations, we determined represented a full community. At this point, the bottom of the grid was over 90% filled and cells throughout the community had begun to divide upwards, generating a three-dimensional structure.

The total number and location of each cell type were stored when a cell in the bottom layer first touched the outer edge. This was intended to capture the community during growth when population structure was forming, but the community was still sparse. Data were also stored at the end of the simulation. In addition to storing the total proportion of each cell type, we stored the proportion of each type at the surface of the colony. To define the colony surface, we began at each point in the top layer of the cubic array and counted downward until we reached the topmost cell. We ran 20 simulations for each combination of parameters; measurements were averaged over all simulations.

As noted above, the growth rate depended on both the social interaction parameters and the neighborhood surrounding the cell. Having a small number of cooperator neighbors increased growth but having too many could reduce growth due to competition. The advantage of increasing the cooperation parameter $r_{SS}$ was thus not directly obvious. By differentiating Eq (1) with respect to occupancy fraction, we found that the maximum growth rate a slow strain could achieve, attained when $\phi_S = \frac{r_{SS} - \chi r_{S0}}{2\chi r_{SS}}$ and $\phi_F = 0$, is $r_{Smax} = \frac{(\chi r_{SS} + r_{S0})^2}{4 r_{SS}\chi}$. In our simulations, we set $\chi = 1$. For the parameter values used here, the maximum slow cell growth rate $r_{Smax}$ was monotonically increasing with the cooperation parameter $r_{SS}$. For easier interpretation of our results, we plotted our simulation results versus the maximum possible growth rate of slow cells rather than the bare parameter $r_{SS}$. Note that the maximum growth rate for baseline competition was 0.05, or the basal growth rate for slow cells. For all other simulations in which the slow cells were cooperators, the maximum growth rate exceeded the basal growth rate.

The simulation code was written in Python and is available on Github (https://github.com/abingham3/SpatialStructureSim); results from this study are available in S2 and S3 Files.

## Results

### Cooperation in a spatially structured community

To investigate the dynamics of the different social scenarios, we began by looking at the proportion of slow cells at the end of the simulations (Fig 2). When compared to the initial proportion in the inoculum, the final proportion can reveal an increase or decrease in frequency of the slow strain throughout growth of the spatially structured community. We also considered the proportion of each cell type on the surface of the community, as dominating the outer edge of an expanding front provides increased access to nutrients and resources, and can be more important for fitness than overall presence throughout the community [36]. We found that proportion of cells on the surface closely mirrored total proportion (Fig 2A), so we present only the total proportion in the rest of the analyses.

In the baseline scenario with no cooperation, and only a fast and a slow growing strain, the clear expectation is that the slow growing strain will be outcompeted by the end of the simulation. This can be seen in the first data point (black square) in each panel in Fig 2A, when $r_{Smax} = 0.05$, which is simply the baseline growth of the slow strain (because $r_{SS} = 0$). In all cases, regardless of the initial proportion of slow cells in the inoculum, the slow strain was nearly non-existent by the end of the simulation.

When cooperation was introduced by allowing slow cells to provide a benefit to other slow cells ($r_{SS} > 0$), as expected, the slow strain remained in the community throughout the simulation and, in some cases, even increased in frequency. The results of obligate cooperation (when slow cells/cooperators provide a benefit to all nearby cells) can be seen in purple in Fig 2. Note that in this scenario, cooperator cells also increased the growth rate of non-cooperator cells, allowing "cheating". When cooperation was very strong ($r_{Smax} = 0.15$ or $0.20$), the cooperator genotype was able to increase in frequency compared to its initial frequency. Thus, these results qualitatively support previous work that showed that clonal growth in a spatially structured community can favor cooperation, even when cheating is possible, as long as public goods are only locally diffused [30, 32].

In the next set of simulations, the slow strain was again a cooperator, but this time was able to restrict its benefit to other cooperators, thus simulating a greenbeard (green points in Fig 2). In this scenario, when the maximum possible growth rate of the cooperator was equal to the baseline growth rate of the non-cooperator ($r_{F0} = 0.1$), the final proportion of cooperators was greater than its initial proportion. This suggests a benefit to a greenbeard recognition system, even in a spatially structured community. Compared to the obligate cooperator scenario, as the initial fraction of slow cells increased (comparing the four panels), the final proportion of slow cells saturated more quickly with increasing maximum growth rate. Overall, the frequency of slow cells was higher when expressing greenbeard cooperation than when non-cooperators were able to benefit from their public goods. When visualizing the resulting communities in the two types of simulations (Fig 2B), a difference in spatial distribution is apparent. Communities with obligate cooperators appear to be more mixed than communities with greenbeard cooperators, which appear to have cooperators spatially separated. This makes intuitive sense, as obligate cooperators are providing benefits to non-cooperators, while greenbeard cooperators are excluding non-cooperators. When a greenbeard is viewed as the ability of cells to adhere to one another, this spatial exclusion supports the observation that adherence can be an advantageous trait in microbial competition [37].

### A comparison of obligate and greenbeard cooperation

To determine how much the slow strain benefited from restricting its cooperation (i.e., expressing a greenbeard recognition signal), we looked at the difference between the total final

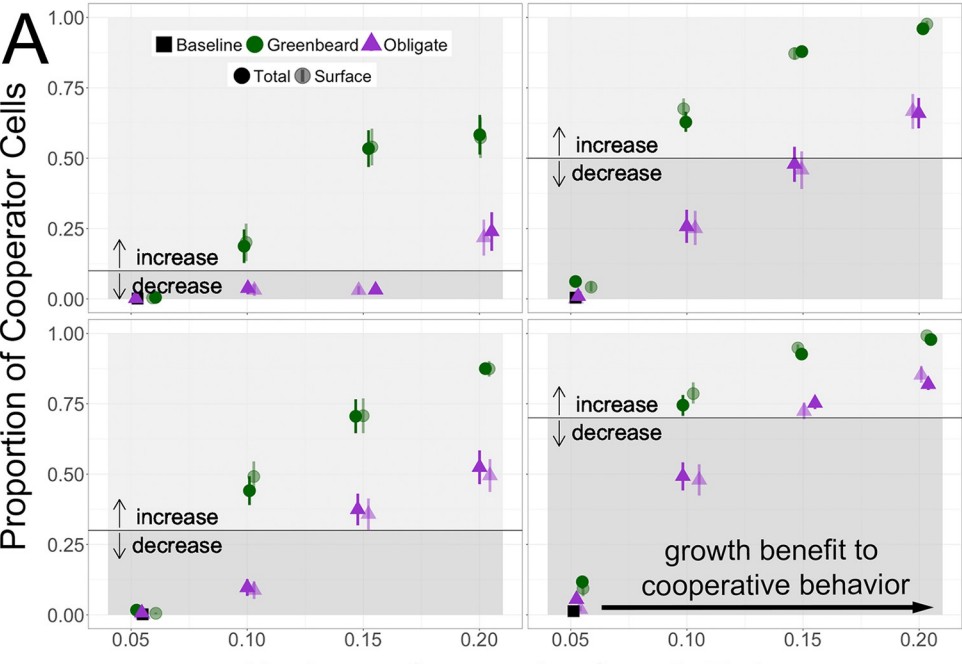

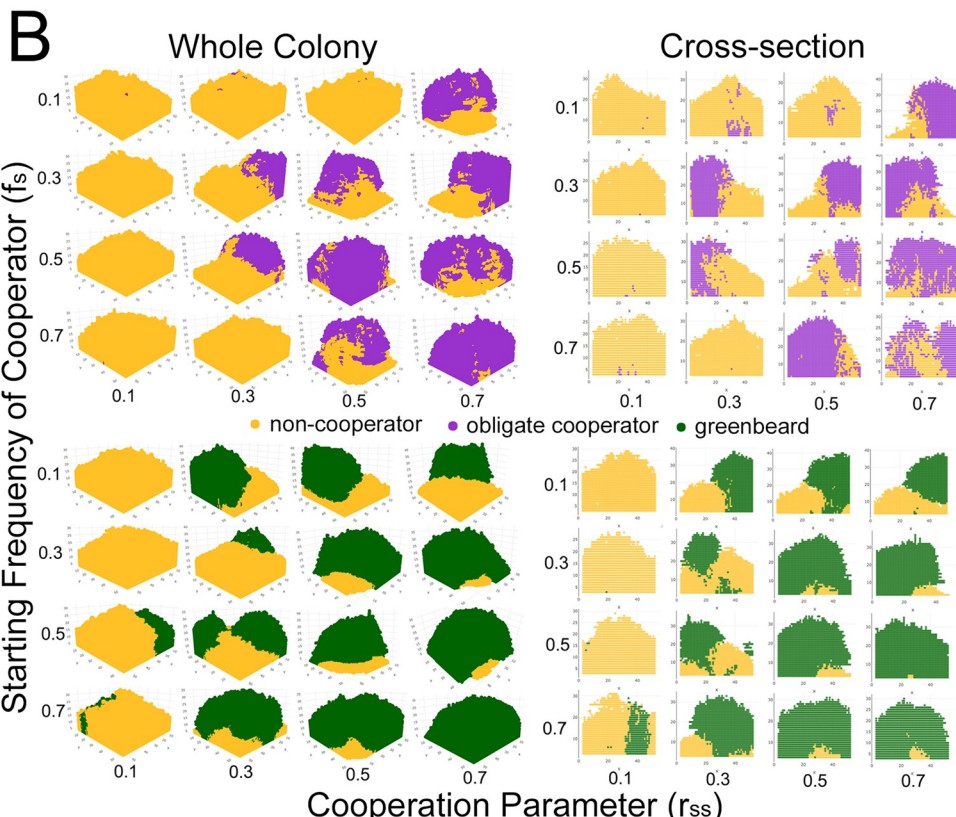

**Fig 2. Results of the greenbeard simulation.** A: Proportion of the slow, cooperator strain at the end of simulations in the total colony and on the outer surface of the colony over a range of cooperative parameter values ($r_{SS}$ = 0.1, 0.3, 0.5, 0.7). Purple points represent obligate cooperation, where the slow strain provides a benefit to all nearby cells. Green points represent greenbeard cooperation, where the slow strain provides a benefit only to other slow cells. Solid points represent the total final proportion of cooperators; transparent points represent the final proportion of cooperators at

the top of the community. Baseline growth rate is the first data point, $r_{Smax}$ = 0.05. Error bars are standard error of the mean from 20 simulated communities. Panels represent a range of initial proportions $f_S$ of the slow strain, indicated by the black lines; arrows represent an increase or decrease of the cooperator strain compared to the initial frequency. Data points are jittered for clarity and are plotted only for the four specified $r_{SS}$ values. B: Sample simulated colonies (left) and cross-sections (right) over a range of cooperative parameter values and cooperator starting frequencies. The first simulated colony for each combination of parameters was plotted in the 3-dimensions used by the simulation ($x$, $y$, and $z$ axes in units of cell length), as well as a 2-dimensional vertical slice from the middle of the colony (at $y$ = 25). Yellow points represent non-cooperators, purple points represent obligate cooperators, and green points represent greenbeard cooperators.

proportion of slow cells in greenbeard and obligate cooperation (Fig 3), i.e., subtracting the purple points in Fig 2 from the corresponding green points. The results suggest that cooperator cells gained more of a benefit from a greenbeard system when they were a moderate fraction (roughly 50%) of the cells present. When there were more cooperator cells (i.e., when $f_S$ = 0.7), there were not as many non-cooperators to take advantage of slow cells, so restricting cooperation had less of an effect. On the other hand, when the cooperators were initially rare, there was a larger benefit to restricting cooperation as long as cooperation was strong enough. Especially when the initial proportion of slow cells was 10%, the benefit to restricting cooperation was low for smaller values of max growth rates.

## Cooperation in sparse versus dense communities

Next, we investigated how the density of the communities affected the benefit of the two different types of cooperation. As shown in the previous two sections, when the communities were fully grown, there was an apparent benefit to obligate cooperation and a further benefit to greenbeard cooperation. This may be in part because the community has grown to a dense enough state for cells to have fuller neighborhoods and the opportunity to interact with one another. To test whether or not the density of the community had an effect on the benefit to cooperation, we compared the proportion of slow cells when the community was sparse and growing exponentially (i.e., when the first cell reached the outer edge of the cubic growth area) with the proportion of slow cells in a dense community at the end of the simulation (i.e., when the community had reached 50,000 cells).

As predicted, in general, slow cells had a higher total proportion at the end of the simulations than in sparse colonies (S1 Fig). This suggests that the growth benefit to both obligate and greenbeard cooperation appear later in the simulation and in more dense conditions. This is likely because in a denser community and after a period of initial growth, slower cooperative cells are surrounded by more of their own kind and are able to benefit from each other.

We compared the benefits of obligate and greenbeard cooperation in sparse and dense conditions (Fig 4) and found that for all starting frequencies of cooperators, the benefit to greenbeard cooperation was generally greater in dense than sparse colonies. This is due to a higher chance of slow cells being near fast cells in denser colonies, and therefore a greater benefit to restricting cooperation and excluding fast cells. We also expect the greatest benefit to greenbeard cooperation when each cell type is present in equal numbers, since there are likely to be more interactions between the types. Indeed, conditions where the final proportion of each cell type is on the order of 50% (see Fig 2B) are associated with the highest benefit to greenbeard cooperation (Fig 4).

## A comparison of greenbeard and polychromatic greenbeard cooperation

The first set of simulations and analyses showed that greenbeard cooperation could increase in frequency more than obligate cooperation under certain conditions in spatially structured

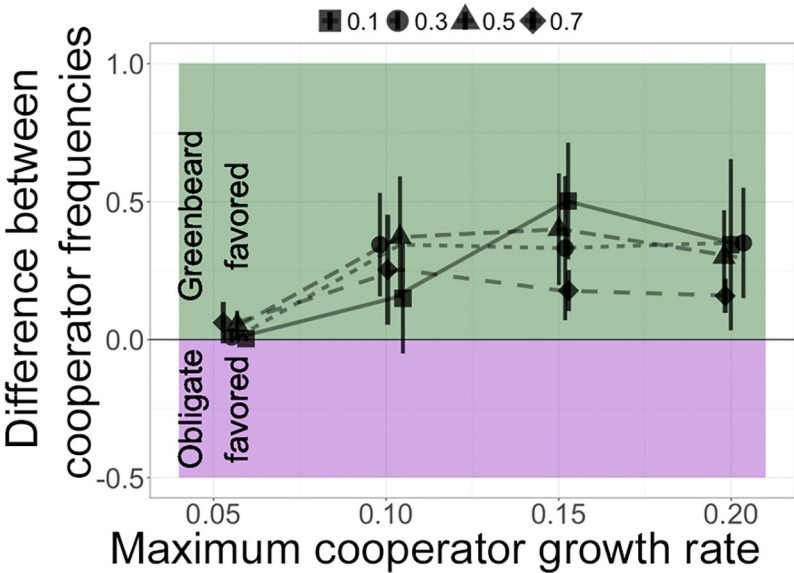

**Fig 3. Growth benefit of cooperation.** Comparison of types of cooperation when in competition with non-cooperators for a range of initial proportions and maximum cooperator cell growth rates. A value of 0 indicates the same final frequency of cooperator cells in obligate and greenbeard cooperation scenarios. Values above 0 indicate a higher final frequency of cooperator cells in the greenbeard scenario; error bars are standard error of the mean. Initial cooperator frequencies: 0.1- square and solid, 0.3- circles and dotted, 0.5- triangles and dashes, 0.7- diamonds and long dashes. Data points are jittered for clarity.

communities. We next sought to determine whether a polychromatic greenbeard (i.e., "true" kin recognition, or recognition of close relatives) could provide a further benefit over that which is provided by a general greenbeard. In natural circumstances, microbial lineages of the same species can have different baseline growth rates due to genetic differences. Even if a community contains only cooperating genotypes, it may benefit faster growing lineages to restrict cooperation in order to not be hampered by the slower growing lineages. This might explain why such a recognition system is observed in real species. To test this hypothesis, we again simulated communities with a fast and a slow growing strain, but this time, both were cooperative and we varied combinations of obligate and restricted cooperation (i.e., simple greenbeard and polychromatic greenbeard).

Fig 5 presents the results of simulations of four scenarios: (1) a fast and slow greenbeard cooperator, (2) a fast greenbeard and a slow polychromatic greenbeard, (3) a fast polychromatic greenbeard with a slow greenbeard, and (4) a fast and slow polychromatic greenbeard cooperator. The scenarios are summarized in Fig 6A. In all cases, we expected the fast strain to ultimately outcompete the slow strain; however, we explored whether certain scenarios enhanced or thwarted this process. A further starting ratio was added, $f_S = 0.9$, in order to clearly see the effect of the social scenarios. In all cases, we set $r_{FF} = r_{SS}$, so both strains produced the same amount or quality of public good.

The baseline of no cooperation can again be seen in the first data point in each panel of Fig 5B ($r_{imax} = r_{i0}$, or baseline growth), where the fast strain has outcompeted the slow strain. Looking at the four social scenarios that include combinations of cooperation, we see that the slow strain decreased in frequency over the timeframe of the simulations, regardless of the starting ratio: its long-term fate is presumably, and not surprisingly, to be outcompeted. However, the slower strain did remain a noticeable fraction in many of the simulations, which can

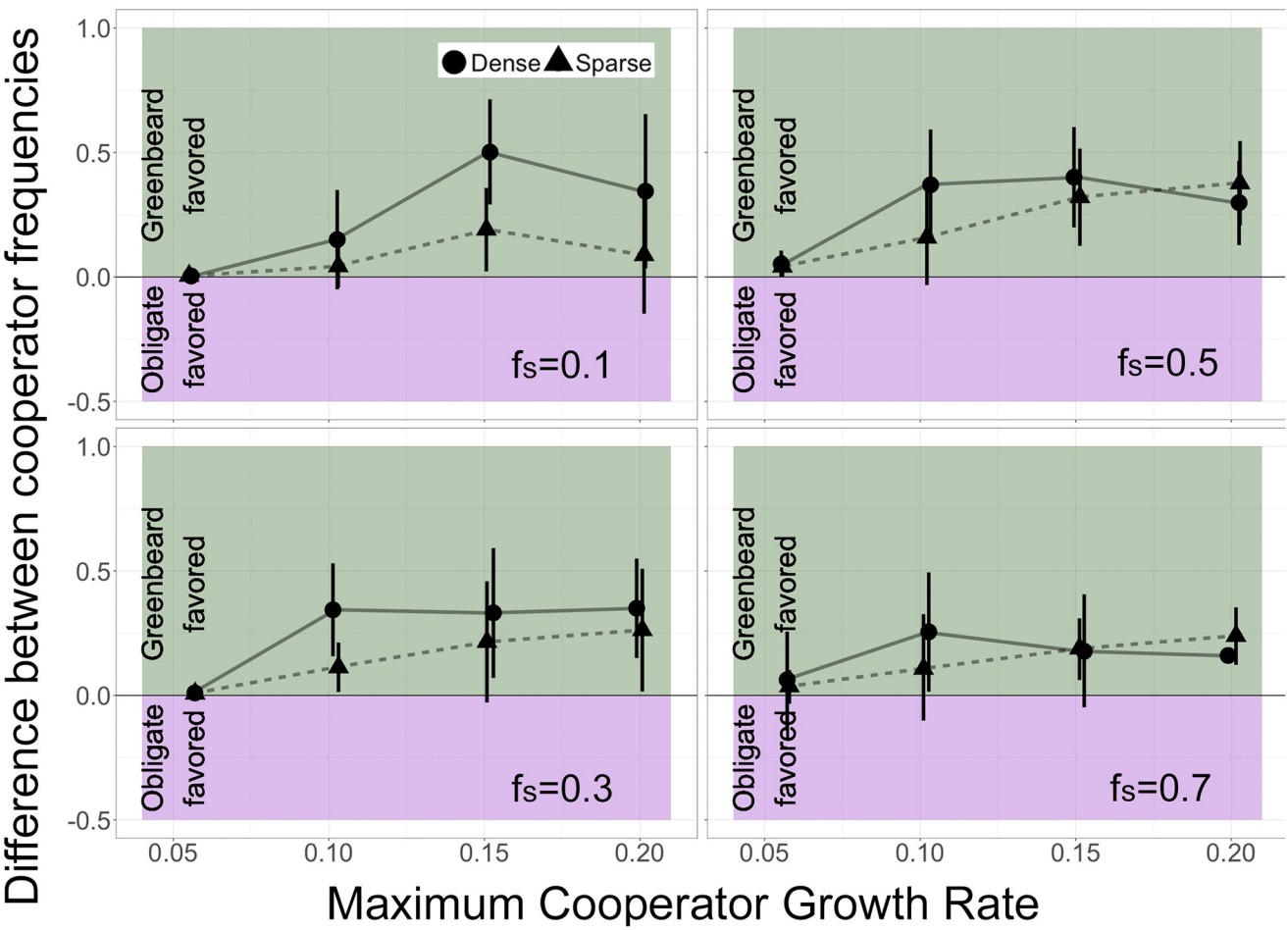

**Fig 4. Sparse vs. dense communities.** Benefit to greenbeard cooperation when colonies are sparse (dashed) or dense (solid). Comparison of types of cooperation when in competition with non-cooperators in sparse (triangles and dashed lines) and dense (circles and solid lines) growing conditions for a range of initial proportions and maximum cooperator cell growth rates. Value interpretation as in Fig 3.

be seen visually in the colony renderings in Fig 5A. As could have been anticipated, in all the scenarios investigated, the results are typically more pronounced the larger the cooperative benefit (i.e., as the maximum possible growth rate from cooperation increased).

We considered, in turn, the benefit to either the slow strain or the fast strain in developing the polychromatic trait, i.e., in restricting its cooperation. The slow strain became polychromatic as it transitioned from scenario 1 to 2 (when competing against a fast greenbeard) or from 3 to 4 (when competing against a fast polychromate). The benefit to the slow strain was greatest when the fast strain was a greenbeard and when cooperation was stronger. Most relevant is the case where the slow strain was initially rare (lower right panel of Figs 5B and 6B), as one can observe whether a new, rare polychromatic mutation competed successfully. We saw that only when the fast strain was a greenbeard that provided substantial cooperation did the slow strain benefit much from becoming polychromatic. Otherwise, the slow strain was too heavily outcompeted by the fast strain.

The fast strain became polychromatic when it transitioned from scenario 1 to 3 (competing against a slow greenbeard) or from 2 to 4 (competing against a slow polychromate). The fast strain was initially rare in the upper left panel of Figs 5B and 6C. Gaining a polychromatic trait

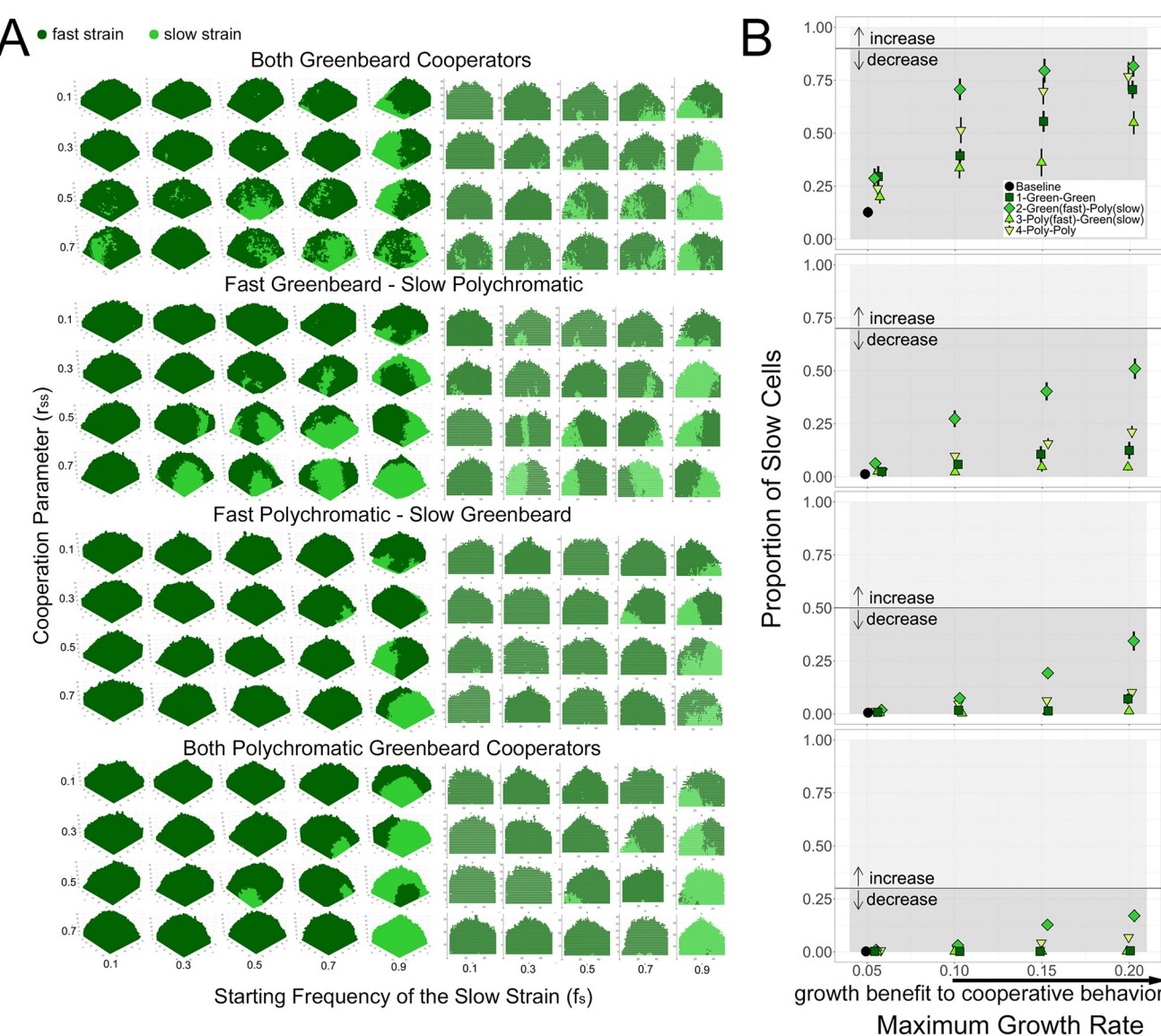

**Fig 5. Cooperator interaction simulations.** A: Sample simulated colonies (left) and cross-sections (right) over a range of cooperative parameter values and starting frequencies. The first simulated colony for each combination of parameters was plotted in the 3-dimensions used by the simulation (*x*, *y*, and *z* axes), as well as a 2-dimensional vertical slice from the middle of the colony (at *y* = 25). Dark green points represent the fast strain and light green points represent the slow strain. B: Total proportion of slow cells at the end of the simulations vs maximum growth rate of the slow strain. In each case, $r_{FF} = r_{SS}$. For Scenario 1, both strains are simple greenbeards (i.e., they cooperate with both cell types). For Scenario 2, the fast strain is a simple greenbeard, but the slow strain is a polychromatic greenbeard (*F* cooperates with *S*, but *S* does not cooperate with *F*). Scenario 3 is the opposite, the fast strain is a polychromatic greenbeard, while the slow strain is simple greenbeard. For the last scenario, both strains are polychromatic greenbeard, so cooperate with their own, but not the other, cell type. The black lines are the initial proportion of the focal strain. Error bars are standard error of the mean; points are jittered for visual clarity.

was beneficial to the fast strain, regardless of whether the slow strain was a greenbeard or polychromatic, and especially when cooperation was stronger. From the subsequent panels in Fig 5B, we see that as the fast strain competed against a slow polychromatic strain, the benefit to the fast strain restricting its cooperation continued to be apparent. However, a slow greenbeard was a poor enough competitor that the fast strain gained less by restricting its cooperation.

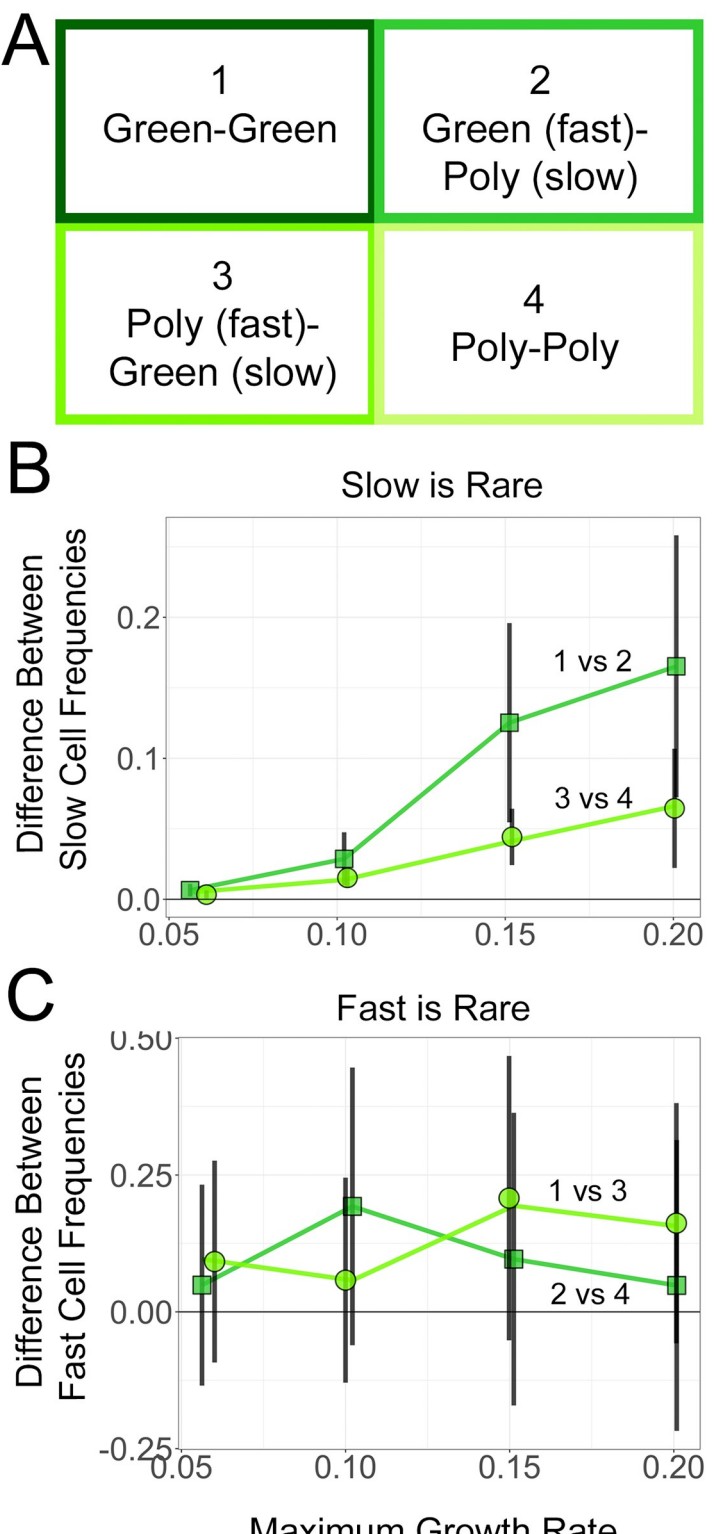

**Fig 6. The effect of restrictive cooperation in cooperative communities.** A: Schematic of cooperative relationships between fast and slow cell types that were simulated in the study. The different competitions are labeled with a number for ease of comparison in panels B and C. B: Comparison of final slow frequencies when slow cells gain restrictive cooperation in simulations begun at 0.3 slow cell frequency. C: Comparison of final fast frequencies when fast cells gain restrictive cooperation in simulations begun at 0.1 fast cell frequency. Error bars are standard error of the mean; data points are jittered for clarity.

In the colony renderings in Fig 5A, the type of cooperation led to some visual differences in the colonies. When both strains were greenbeards, the two strains appeared more intermixed. When both were polychromatic, each strain appeared more clustered with its own kind. However, the intermixing in the two-greenbeard case was probably less than in the double-cooperator scenario of [35] because our greenbeards cooperated with their own cell type in addition to the other type, obviating the need for each cell to grow near cells of the other type. The effect of cooperation type on colony structure would be an interesting topic for further study.

## Discussion

The goal of the simulations presented here was to investigate the effects of different types of cooperation on the frequency and arrangement of cells in spatially structured microbial communities, and in particular, determine whether a polychromatic greenbeard (i.e., microbial kin recognition) could provide a fitness or growth benefit in such a community. Our simulations began with a small inoculum of different starting frequencies of two cell types. This could represent an experimental situation, such as starting a colony on a petri dish, or a more natural situation, such as a migration via an insect to a new environment or resource patch. The probability of growth for each cell depended on its type and neighborhood. We tracked the colony growth over time and monitored the change in frequency of the cell types. The simulations were fitness-based models that altered growth rates based on different types of cell-cell interactions. The model did not consider biological mechanisms, such as nutrient flux or enzyme diffusion, as the goal was to determine whether certain types of interactions could provide a fitness benefit, not to model the specific dynamics of an experimental system [29, 38].

Despite not including explicit mechanisms in our model, the scenarios we simulated are rooted in biological phenomena that exist in spatially structured communities. The obligate cooperators growing with non-cooperators captures the idea of a locally diffusible enzyme that can be used by producers and non-producers alike (e.g., [39–43]), or floating aggregates that can contain non-producing cells that ultimately lead to the collapse of the community [44]. It is within this context that it is possible to ask whether a recognition mechanism in which cooperators only provide growth benefits to other cooperators might provide a fitness advantage, which was tested in our greenbeard and non-cooperator simulations. For biofilms, these simulations capture the idea of only adhering to cells also expressing adhesins [17, 19]. And finally, we ask if restrictive cooperation in which cooperators provide growth benefits exclusively to specific cooperator types might provide a further fitness advantage, tested in our polychromatic greenbeard simulations. In the case of diffusible enzymes, this could be the advantage of enzyme-receptor specificity [45], while in the case of biofilms, it could be adhering to cells with the same adhesin alleles [34].

In the first set of simulations, our results confirmed that obligate cooperation can increase in frequency in spatially structured communities, even when "cheating" [3, 46] is possible, as long as the growth benefit provided by cooperative cells is only local. This has been studied most extensively as diffusible public goods using experimental, computational, and theoretical approaches (reviewed in [29]). Our simulation model results qualitatively agree with the conclusions of those studies: when growth promotes clonal clusters, the benefit to the cooperative behavior is mostly shared with clonemates, and can therefore resolve the public goods cheating dilemma.

In this first set of simulations, we also investigated greenbeard cooperation in a community with non-cooperators. In microbes, this kind of cooperation is usually in the form of cell-cell adherence, and has been investigated experimentally and computationally [17, 19, 30, 32, 37,

47]. Once again, our simulations qualitatively confirm findings from these studies: the ability to adhere selectively promotes spatial segregation and can even be used as a weapon against non-adhering strains.

In our simulations, we were able to compare the increase in frequency of cooperators under the different types of cooperation, and found that greenbeard cooperation is able to increase in frequency more than obligate cooperation, a somewhat surprising result for spatially structured communities. The clonal growth of stationary cells passively generates local clusters, so it was unclear in advance whether a greenbeard cooperator would provide much of a growth advantage compared to an obligate cooperator that is able to interact within a cluster of like types. Finally, we found that the benefit to both obligate and greenbeard cooperation when grown with non-cooperators was realized as communities became denser at the end of the community expansion. The increase in density allowed the growth benefits of cooperators to have an effect as cells were in contact more [48].

Our second set of simulations investigated whether a polychromatic greenbeard, in which the cells only cooperate with their own type, could provide a numerical benefit beyond a simple, binary greenbeard. The results showed that a more fit, faster-growing lineage will be hampered by the presence of a less fit, slower-growing lineage because the slower lineage will gain a growth benefit from being near the more abundant, faster one. The results also show that if one of the strains were able to restrict their cooperation (the asymmetric case of one polychromatic greenbeard and one simple greenbeard), such a strategy would be favored over the symmetric greenbeard case (i.e., the faster strain increases its relative frequency when it is selectively cooperative, and the slower strain increases its relative frequency when it is selectively cooperative). Finally, we also see that restoring the symmetry, by both strains only cooperating with their own type, would be favored. That is, when one strain has the advantage of restricting cooperation (i.e., a form of "cheating"), if other strain also becomes selectively cooperative, its frequency will increase. This suggests that if one lineage were to evolve a mechanism for selective recognition, selection would favor a similar mechanism in the other. This could lead to a polychromatic greenbeard cooperative system in a spatially structured community. The question of the general conditions that favor the evolution and stability of kin recognition is different than that which is addressed here (reviewed in [49]). We simply show that there are conditions under which a recognition system could provide a numerical fitness benefit.

The simulation model used for this research was inspired by an investigation into the spatial arrangement of microbial communities with different types of ecological interactions [35]. While the goal of our work was to compare the effects of obligate and restrictive cooperation on population growth, and not to measure spatial characteristics per se, it is worth noting the similarities and differences between the sets of simulated communities from the two research endeavors. Briefly, Momeni et al. described the effect of one cell type on the other as either neutral ($\sim$), positive ($\uparrow$), or negative ($\downarrow$), but assumed the effect of a cell on its own type was neutral. For example: ($\sim$, $\sim$) was baseline cooperation in which cell types simply competed for resources; ($\sim$, $\downarrow$) was amensalism in which one cell type harmed the other; and ($\uparrow$, $\uparrow$) was cooperation in which each cell type benefitted from the other.

A main conclusion from Momeni et al.'s research was that cooperation ($\uparrow$, $\uparrow$) led to spatial "intermixing" of lineages, while other ecological interactions led to spatial segregation. In their conceptual framework, cooperation was defined as a relationship *between* lineages and was inspired by the biological phenomenon of two auxotrophic cell types secreting products that complement the other's needs. It therefore makes intuitive sense for the cell lineages to mix as they enhance each other's growth. In our simulations, cooperation was defined *within* a lineage and was inspired by a different biological phenomenon, one in which a single lineage produces goods that enhance its own growth, but goods that can possibly be utilized by others, unless

restricted. Comparing the models, our "obligate cooperation" (with the potential for non-cooperators to be considered cheaters) is the same as their commensal ($\sim$, $\uparrow$), except in our case, the lineage providing the growth benefit also helps its own growth. Similarly, our "two simple greenbeards" is the same as their ($\uparrow$, $\uparrow$) cooperation, except again, our lineages also enhance their own growth.

We did not calculate an intermixing index, but we can ask qualitatively, what kind of spatial arrangements were generated by the cooperative interactions in our simulations? Most of our simulated colonies showed strong spatial segregation. The two scenarios that showed the most intermixing between lineages were: (1) an obligate cooperator with a non-cooperator (Fig 2B), and (2) two simple greenbeards (Fig 5A); however, neither of these scenarios appeared to show as much mixing as the ($\uparrow$, $\uparrow$) cooperation in the Momeni et al. simulations. Notably, these are the only two cases in which none of the lineages can restrict the growth benefits of cooperation. Using their definitions, restricting cooperation simply reverts a lineage to neutral ($\sim$) in its effect on the other strain. Thus, most of our simulations fall within the "non-cooperation" designation of their framework and have the spatial assortment associated with it. It thus appears that the amount of intermixing expected in a cooperative microbial community depends strongly on the type of cooperation: restricting the benefits of cooperation to like kinds can lead to enhanced segregation, rather than the cooperative intermixing previously reported.

While the details of the simulations and the parameter values used here may not be representative of all, or even most, microbial communities, the qualitative result, that there are possible scenarios that favor a form of "kin recognition" in spatially structured microbial communities, may be more general. They may also shed light on recent observations in yeast and bacteria that suggest specificity in social traits may exist in natural microbial populations [34, 45]. In a study of *Pseudomonas* bacteria, producers and non-producers of pyoverdine (an iron-scavenging compound considered to be a local public good) were isolated within soil communities, suggesting cooperators and cheaters exist together in nature in a spatially structured environment [45]. Furthermore, the ability of non-producers to use the pyoverdine of producers was affected by relatedness. In the *Saccharomyces* yeasts that inspired this research, different genotypes can be found as close as on the same tree [50, 51] and research with *Drosophila* has shown that yeast spores can traverse the digestive tract with viable spores deposited post-digestion [52]. Cellular adhesins required for yeast biofilm formation show different adherence properties in self and non-self combinations [34]. Thus, there are plausible ecological scenarios in which multiple genotypes may inoculate and compete in a new environment. More broadly, in many natural microbial populations, diverse lineages are isolated from the environment, and the diversity is often related to the ecological function and evolutionary history of the species [53]. Thus, the potential exists for the type of interactions studied here to exist in natural settings.

## Supporting information

**S1 Fig. Comparison of sparse and dense colonies in different cooperator scenarios.** Total proportion of slow strain (*S*) when colonies are sparse (light) or dense (solid). Strain *S* are cooperators and strain *F* are non-cooperators; purple triangles represent obligate cooperation and green circles represent greenbeard cooperation. The horizontal axis is the maximum growth rate for strain *S*. Each panel represents a different initial proportion of *S*. Error bars are standard error of the mean.
(TIF)

**S1 File. Simulation details.** Description of growth, competition, and cooperation in the simulations.
(PDF)

**S2 File. Simulation results.** Summary of cell proportions in the total and at the edge of the communities for the obligate and greenbeard simulations.
(CSV)

**S3 File. Simulation results.** Summary of cell proportions in the total communities for the polychromatic greenbeard simulations.
(CSV)

## Acknowledgments

We thank William & Mary Research Computing for providing computational resources that have contributed to the results reported within this paper.

## Author Contributions

**Conceptualization:** Adrienna Bingham, Leah B. Shaw, Helen A. Murphy.

**Data curation:** Adrienna Bingham.

**Formal analysis:** Adrienna Bingham, Aparajita Sur.

**Funding acquisition:** Helen A. Murphy.

**Investigation:** Adrienna Bingham, Aparajita Sur, Leah B. Shaw.

**Methodology:** Adrienna Bingham, Aparajita Sur.

**Project administration:** Leah B. Shaw, Helen A. Murphy.

**Writing – original draft:** Adrienna Bingham, Helen A. Murphy.

**Writing – review & editing:** Leah B. Shaw, Helen A. Murphy.

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
