## [Decision Letter · Decision Letter 0]

12 Oct 2023

PONE-D-23-24257The effect of cooperator recognition on competition among clones in spatially structured microbial communitiesPLOS ONE

Dear Dr. Murphy,

Thank you for submitting your manuscript to PLOS ONE. After careful consideration, we feel that it has merit but does not fully meet PLOS ONE’s publication criteria as it currently stands. Therefore, we invite you to submit a revised version of the manuscript that addresses the points raised during the review process.

Thank you for submitting your work to PLoS One.

Two reviewers have reviewed your original submission. Both reviewers indicated a minor revision was required. Reviewer one would like more to see more information on the modeling methods (and a minor fix to a figure legend) while reviewer 2 would like you to address how the simulations connect with natural systems. To me, the latter could be done in the discussion and I believe that this is what the reviewer is alluding to. I look forward to reading your revised manuscript.

We look forward to receiving your revised manuscript.

Kind regards,

Robert P Smith

Academic Editor

PLOS ONE

Reviewers' comments:

Reviewer's Responses to Questions

**Comments to the Author**

1. Is the manuscript technically sound, and do the data support the conclusions?

Reviewer #1: Yes

Reviewer #2: Yes

2. Has the statistical analysis been performed appropriately and rigorously? 

Reviewer #1: I Don't Know

Reviewer #2: Yes

3. Have the authors made all data underlying the findings in their manuscript fully available?

Reviewer #1: No

Reviewer #2: Yes

4. Is the manuscript presented in an intelligible fashion and written in standard English?

Reviewer #1: Yes

Reviewer #2: Yes

5. Review Comments to the Author

Reviewer #1: The manuscript titled "The effect of cooperator recognition on competition among clones in spatially structured microbial communities" describes an agent based modeling approach where cooperation in biofilm is assessed. An important feature is that the models incorporate kin recognition/selection and by doing so the can model the effects of it in spatially structured environments. The tested conditions are biologically relevant.

The paper is clearly written and the results make intuitive sense. The authors are clear about the limitations of the study.

As the models are not shared and I am not an expert on these type of models I can not assess those details.

While the general concepts are well described in the methods section it is not mentioned what software was used for the modeling and I think the models itself should be made available as well.

Figs 1,2,5: From the legend it is not clear what is on the x,y and z axis of the 3D plot

The legend of Fig 6 needs to be extended. In the current form I do not understand what the authors are trying to say with Figure 6a for instance and how it links to the other panels.

Reviewer #2: This study investigated the effects of different types of cooperation in a microbial colony using spatially explicit, agent-based simulations of two interacting strains. They found scenarios that favor a form of kin recognition in spatially structured microbial communities. The observed results suggest that in some ecological scenarios, recognition mechanisms may be beneficial even in spatially structured communities. This study is interesting and well organized. My major concern is on how to build relationships between this simulation results to ecological systems. As we know the environment selects the microorganisms. So how to determine the different roles of environmental conditions and microbial interactions on microbial community development. Please discuss these contents in the manuscript. I think it could be accepted by minor revisions.

6. PLOS authors have the option to publish the peer review history of their article (what does this mean?). If published, this will include your full peer review and any attached files.

Reviewer #1: No

Reviewer #2: No

---

## [Author Response · Author response to Decision Letter 0]

1 Feb 2024

Editor’s Summary:

Two reviewers have reviewed your original submission. Both reviewers indicated a minor revision was required. Reviewer one would like more to see more information on the modeling methods (and a minor fix to a figure legend) while reviewer 2 would like you to address how the simulations connect with natural systems. To me, the latter could be done in the discussion and I believe that this is what the reviewer is alluding to. I look forward to reading your revised manuscript.

>To the best of our ability, we have the made the revision requested by both reviewers and hope the manuscript is now suitable for publicatioin.

Reviewer #1: 

(1) The manuscript titled "The effect of cooperator recognition on competition among clones in spatially structured microbial communities" describes an agent based modeling approach where cooperation in biofilm is assessed. An important feature is that the models incorporate kin recognition/selection and by doing so the can model the effects of it in spatially structured environments. The tested conditions are biologically relevant.

The paper is clearly written and the results make intuitive sense. The authors are clear about the limitations of the study.

As the models are not shared and I am not an expert on these type of models I can not assess those details.

While the general concepts are well described in the methods section it is not mentioned what software was used for the modeling and I think the models itself should be made available as well.

>We have added a line in the Methods that notes the simulations were written in Python and have made the code available through GitHub. We have done our best to comment extensively in the code.

(2) Figs 1,2,5: From the legend it is not clear what is on the x,y and z axis of the 3D plot

>The figures with colonies represent the simulations. The axes provide the location for the cells in the simulated cubic array and are in units of cell length. We have updated the text to clarify this in Figures 1 and 2.

(3) The legend of Fig 6 needs to be extended. In the current form I do not understand what the authors are trying to say with Figure 6a for instance and how it links to the other panels.

>Figure 6a is showing the different strategies being competed against each other and should be used as a key for interpreting Figure 6b. We have updated the legend to clarify.

Reviewer #2: 

This study investigated the effects of different types of cooperation in a microbial colony using spatially explicit, agent-based simulations of two interacting strains. They found scenarios that favor a form of kin recognition in spatially structured microbial communities. The observed results suggest that in some ecological scenarios, recognition mechanisms may be beneficial even in spatially structured communities. This study is interesting and well organized. My major concern is on how to build relationships between this simulation results to ecological systems. As we know the environment selects the microorganisms. So how to determine the different roles of environmental conditions and microbial interactions on microbial community development. Please discuss these contents in the manuscript. I think it could be accepted by minor revisions.

>We appreciate the encouragement to connect our simulation and its results to the actual biology of microorganisms. It is absolutely true that the environment selects the organism. However, when multiple lineages of a single species find themselves together in an environment conducive to growth, they will be interacting with each other. We are not sure we can answer the question of how to determine the role of the environment vs. social interactions in structuring microbial communities, but we have updated the end of the discussion to provide plausible biological scenarios in which multiple lineages of one species could find themselves growing and interacting.

---

## [Decision Letter · Decision Letter 1]

13 Feb 2024

The effect of cooperator recognition on competition among clones in spatially structured microbial communities

PONE-D-23-24257R1

Dear Dr. Murphy,

We’re pleased to inform you that your manuscript has been judged scientifically suitable for publication and will be formally accepted for publication once it meets all outstanding technical requirements.

Kind regards,

Robert P Smith

Academic Editor

PLOS ONE

Additional Editor Comments (optional):

Thank you for addressing all of the reviewer comments.

Reviewers' comments:

Reviewer's Responses to Questions

**Comments to the Author**

1. If the authors have adequately addressed your comments raised in a previous round of review and you feel that this manuscript is now acceptable for publication, you may indicate that here to bypass the “Comments to the Author” section, enter your conflict of interest statement in the “Confidential to Editor” section, and submit your "Accept" recommendation.

Reviewer #2: All comments have been addressed

2. Is the manuscript technically sound, and do the data support the conclusions?

Reviewer #2: Yes

3. Has the statistical analysis been performed appropriately and rigorously? 

Reviewer #2: Yes

4. Have the authors made all data underlying the findings in their manuscript fully available?

Reviewer #2: Yes

5. Is the manuscript presented in an intelligible fashion and written in standard English?

Reviewer #2: Yes

6. Review Comments to the Author

Reviewer #2: After I checked the revised manuscript, I think the authors have addressed my concerns. Now I agree the paper can be accepted.

7. PLOS authors have the option to publish the peer review history of their article (what does this mean?). If published, this will include your full peer review and any attached files.

Reviewer #2: No

---

## [Editor Report · Acceptance letter]

17 Mar 2024

PONE-D-23-24257R1 

PLOS ONE

Dear Dr. Murphy, 

I'm pleased to inform you that your manuscript has been deemed suitable for publication in PLOS ONE. Congratulations! Your manuscript is now being handed over to our production team.

Kind regards, 

on behalf of

Dr. Robert P Smith 

Academic Editor

PLOS ONE